# Fertilizer-Holding Performance of Graphene on Soil Colloids Based on Double Electric Layer Theory

**DOI:** 10.3390/ma16072578

**Published:** 2023-03-24

**Authors:** Ziyan Liu, Ming Zhou, Wufang Liao, Jiayi Liu, Chaogui Luo, Chunyan Lu, Zhiwen Chen, Hongwei Zhu

**Affiliations:** 1School of Mechanical and Automotive Engineering, Guangxi University of Science and Technology, Liuzhou 545006, China; 2Guangxi Earthmoving Machinery Collaborative Innovation Center, Liuzhou 545006, China; 3Guangxi Tsinglube New Material Technology Co., Ltd., 279 Feilu Avenue, Luzhai County, Liuzhou 545006, China; 4Agricultural Technology Extension Center in Luzhai County, Liuzhou 545006, China; 5Guangxi Agricultural Technology Extension Station, Nanning 530001, China; 6Laibin Xingbin District Agriculture Technology Popularizing Station, Laibin 546100, China; 7Hainan Yazhou Bay Seed Laboratory, Sanya 572025, China; 8National Key Laboratory of Plant Molecular Genetics, National Center for Plant Gene Research, Institute of Plant Physiology and Ecology/CAS Center for Excellence in Molecular Plant Sciences, Chinese Academy of Sciences, Shanghai 200032, China; 9State Key Lab of New Ceramics and Fine Processing, School of Materials Science and Engineering, Tsinghua University, Beijing 100084, China

**Keywords:** double electric layer, graphene, nutrients, soil colloids, zeta potential

## Abstract

Soil nutrient loss, which leads to low plant utilization, has become an urgent issue. Graphene can improve soil fertilizer-holding properties given its small size effect, strong adsorption properties, and large specific surface area. Herein, different amounts of graphene were added to soil samples to study its effect on soil nutrient retention and growth of pepper seedlings. The colloidal double electric layer theory forms the basis for an analysis of variations in soil nutrient concentration through measurements of the zeta potential, which is affected by variations in ion concentrations in soil colloids. We measured the zeta potential of graphene and soil mixed colloids and found that graphene could increase the concentration of nutrient ions in soil colloids. In addition, graphene reduced the loss of nutrients; increased the contents of ammonium nitrogen, effective phosphorus, and fast-acting potassium in the soil after leaching; and enhanced the stability of soil aggregates after leaching. In addition, pepper seedlings grown under graphene treatment for 60 days outperformed seedlings grown without graphene treatment, in terms of plant height and nutrient content. This study demonstrates that the addition of graphene to soil can reduce nutrient loss and promote fertility and plant growth.

## 1. Introduction

Contact between solid and liquid phases leads to the formation of a double-layer structure because of the differences in their physical properties and/or the ionization of active groups and charge separation at the interface. The double layer theory was initially introduced by Helmholtz in the 19th century, in which the surface charge and counter ions of charged bodies are compared to a flat capacitor. Since then, the double layer theory has been further developed [1,2,3,4]. For example, solid–liquid interfaces have become an important part of interfacial science, and the double electric layer and zeta potential are important components of interfacial chemistry influenced by adsorption and exchange at solid–liquid interfaces. The double electric layer theory has been explored in the fields of chemistry [5], energy [6], and soils [7,8].

Colloidal bilayers are another important aspect of interfacial chemistry [9]. The double electric layer of soil colloids comprises two layers of oppositely charged ions formed at the periphery of the colloidal particles by the electrostatic action and thermal movement of the ions, which affect the physical and chemical properties of the soil. The double electric layer theory has been used to help explain the moisture content and water migration of permafrost [10,11,12,13]. Moayedi et al. [14] investigated the effect of electrolytes on the surface charge and double electric layer thickness in an effort to explain the flocculation and coagulation behavior of peat colloids. Zou et al. [15] studied the adsorption of copper ions by variably charged soils and particles, and when the ion concentration increased, the adsorption of copper ions by the soil particles also increased. This was related to the interaction of two colloidal double electric layers. The double electric layer theory can be applied not only to the analysis of soil structural properties, but can also be used to explain the sorption of nutrient ions to soil particles.

Soil colloids are the most active parts of soil and the main carriers of stored nutrients that plants require. The surface of soil colloids is charged, and this charged nature induces ion adsorption. The greater the colloidal charge, the stronger the adsorption properties. The stability of soil colloids reflects the size magnitude of the charge of the colloidal particles. The greater the charge, the more stable the colloid; that is, the greater the concentration of ions in the soil colloid, the higher the fertility of the soil. Understanding the colloidal double electric layer theory as it applies to soil is important for the study of ion adsorption and exchange at solid–liquid interfaces, nutrient transport at the root–soil interface, and soil composition [16,17,18]. It also provides new insights for the study of changes in soil ion concentration.

Soil nutrient loss and low utilization have become pressing issues requiring new ideas and approaches, including the implementation of functional materials. Graphene, a novel carbon nanomaterial, was produced at a high quality by Novoselov et al. at the University of Manchester using the exfoliative graphite method [19]. The material possesses large π-π conjugate bonds on its surface and a two-dimensional carbon network structure with carbon atoms in a hexagonal honeycomb shape. Compared with other nanocarbons, graphene has a better size effect, a larger specific surface area, and is more stable, making this material effective for soil fertilization applications. Sui et al. [20] found that graphene suspensions could reduce the loss of nitrogen, phosphorus, and potassium from soil through simulated rainfall experiments in soil columns, and the degree of effect increased with an increase in graphene soil concentration. Pandorf et al. [21] found that the application of graphite granules to soil reduced the need for fertilizer application, improved fertilizer utilization, and reduced nitrogen fertilizer losses without reducing the yield of lettuce. Thus, it is clear that graphene has a considerable retention effect on soil nutrients and a fertilizer retention effect, and shows promise for application in soil improvement efforts.

Few studies have focused on using the double electric layer theory to help analyze the effect of graphene on soil colloid stability and soil ion concentration. Changes in the double electric layer of soil are important indicators of changes in ion concentration in soil colloids, facilitating in-depth analysis of ion concentration changes from a microscopic perspective. Furthermore, measuring the zeta potential is useful for characterizing changes in colloid charge. In this study, the effect of graphene on nutrient ion concentrations was analyzed by measuring the zeta potential of soil colloids and applying the Derjaguin–Landau–Verway–Overbeek theory of colloid stability using a double electric layer. We established a leaching experiment to investigate the effect of added graphene on nutrient loss in soil and the fertility retention effect. Furthermore, using planting peppers experiments, we tested the effect of graphene on promoting plant growth and nutrient uptake through its effect on soil fertility. Our findings provide new insights into the application of graphene in agriculture.

## 2. Materials and Methods

### 2.1. Materials and Instruments

Graphene was purchased from Guangxi Qinglu New Material Technology Co., Ltd. (Liuzhou, China). Water-soluble fertilizer (N:P:K, 19:19:19) was purchased from Guangxi Gamma Biotechnology Co. (Nanning, China) Soil was originally obtained from a common garden in Jiangsu Province. The basic physical and chemical properties of the soil were as follows: pH 7.6, ammonium nitrogen content 8.0 mg kg^−1^, effective phosphorus content 6.4 mg kg^−1^, and fast-acting potassium content 149.4 mg kg^−1^. A nanoparticle zeta potentiometer (Horiba nanoPartica SZ-100V2; Horiba, Kyoto, Japan) was used to measure the zeta potential value of soil colloids. A soil fertilizer nutrient detector (HM-TYC, Shandong Hengmei Electronic Technology Co., Weifang, China) was used to measure soil NPK elements.

### 2.2. Test Method

#### 2.2.1. Soil Colloid Zeta Potential Measurement

Soil samples were air-dried, ground, and filtered through a 60-mesh sieve, and colloidal particles <2 µm were extracted using the sedimentation method for zeta potential determination as follows. Four 50 mL beakers were each filled with 10 g of dry soil sample. Subsequently, 4 mL of graphene water dispersion solution (0, 20, 50, and 100 mg L^−1^) and 20 mL of deionized water were added. A glass rod was used to stir the mixture. Following 30 min of ultrasonic soundwave treatment, the mixture was allowed to stand for 1 min, and the upper suspension was collected. The zeta potential of the suspension was measured using the nanoparticle potentiometer. During measurement, the colloidal suspension was moved to an electric pool, and a voltage of 3.3 V was applied at both ends of the electrode (temperature, 25 ± 0.5 °C). Each sample was tested nine times, and the average was calculated to obtain the final averaged zeta potential absolute value.

#### 2.2.2. Leaching Test

Four drenching devices were set up, each divided into three parts: an upper water injection system, a middle soil column, and a lower device for receiving the drenching solution. The water injection system consisted of a 500 mL titration device with an adjustable flow rate, and the soil column consisted of a transparent tube with an inner diameter of 6 cm and a height of 30 cm, with a rubber stopper at one end and a 2 cm diameter round hole in the middle. A layer of filter paper was positioned at the bottom of the column to prevent soil particles from being lost with water during the drenching process, and a layer of filter paper was positioned at the top of the column to prevent damage to the surface soil layer and to facilitate the uniform penetration of water into the soil. Soil was slowly added to a funnel device so that it was evenly spread, and the edges were compacted to avoid the water infiltration at the wall. In each of four experimental setups, 500 g of dry soil was weighed, and then 0.2 g (equivalent to 60 kg/hm^2^ of fertilizer application) of a balanced water-soluble fertilizer, and graphene at concentrations of 0, 20, 50, and 100 mg L^−1^ were added to 500 mL of deionized water, and the samples were denoted as G0, G20, G50, and G100, respectively. After 30 min of ultrasonic dispersion, the graphene dispersed suspensions were slowly and uniformly added to the soil column using a titration device over a period of 5 h. The collected filtrate was immediately used to measure conductivity, and the soil was naturally dried, mixed, and passed through a 60-mesh sieve to determine the content of ammonium nitrogen, effective phosphorus, and fast-acting potassium in the soil. Each experimental group was set up with three replicates; each was measured five times, and the average value was determined in each case.

#### 2.2.3. Soil Agglomerate Stability Measurement

The composition of soil microagglomerates was determined based on the national standard NY/T 1121.20-2008. We weighed 30 g of an air-dried soil sample after over-drenching in a beaker, added 150 mL of distilled water, and soaked the sample for 24 h. The suspension was stirred at 200 r/min for 15 min and passed through a 60-mesh sieve (0.25 mm), before being washed with 1000 mL distilled water in a 1000 mL beaker. Then, we examined particles <0.02 mm, <0.015 mm, <0.005 mm at 10 cm of sinking according to the temperature table and settling time. A siphon tube was gently inserted into the beaker at the settling time, and the colloidal suspension was aspirated into another container before being placed in an oven at 60 °C to dry it prior to weighing.

Soil agglomerate stability indicators were described by the mean weight diameter (MWD) and geometric mean diameter (GMD) [22], which are expressed by the following equations:$$MWD=\sum_{i=1}^{n}W_{i}X_{i}$$
$$GWD=\exp[(\sum_{i=1}^{n}W_{i}\ln X_{i})/\sum_{i=1}^{n}W_{i}]$$
where *n* is the number of groups of particle sizes, *X_i_* is the average diameter of agglomerates in any particle size range (mm), and *W_i_* is the weight percentage of agglomerates in any particle size range (%).

#### 2.2.4. Pepper Seedling Cultivation

For each experimental group, 1 kg of dry soil was placed into planting pots (14 cm diameter × 12 cm height). Forty pepper seeds with full grains and a uniform size were sown equally in the four pots, and graphene was then added at concentrations of 0, 20, 50, and 100 mg L^−1^ dispersed suspensions. After seed germination, 100 mL of water was applied every 5 d, and water-soluble fertilizer was applied every 15 d at a rate of 60 kg hm^−2^. The experiment was replicated three times in each group at 26 ± 1 °C with 8 h of light provided daily. After 60 d of incubation, five plants were randomly selected from each experimental group, the height of pepper seedlings measured, and then dried to measure the content of ammonium nitrogen, effective phosphorus, and fast-acting potassium in the plants.

#### 2.2.5. Nutrient Test Method

The conductivity of the drench solution was measured using a nanoparticle zeta potential meter with 3.3 V applied at both ends of the electrode, and the temperature set at 25 ± 0.5 °C. The content of soil ammonium nitrogen was determined using colorimetry and phenol–hypochlorite as follows. Dry soil (1 g) was treated with potassium sulfate solution, phenol solution, and sodium hypochlorite alkaline solution, and elemental nitrogen was determined using a Kjeldahl nitrogen tester [23]. The effective soil phosphorus content was determined using a molybdenum–antimony anti-colorimetric method and sodium bicarbonate extraction as follows. Dry soil (1 g) was treated with phosphorus-free activated carbon powder, sodium bicarbonate, dinitrophenol indicator, dilute hydrochloric acid, sodium hydroxide solution, and molybdenum–antimony colorant, and elemental phosphorus was determined using a spectrophotometer [23]. The fast-acting soil potassium content was determined using a turbidimetric method, a NaNO_3_ leaching agent, and sodium tetraphenylboron as follows. Dry soil (1 g) was treated with sodium nitrate, CH_2_O-EDTA masking agent, and sodium tetraphenylborate solution, and the elemental potassium was measured by flame photometry [23,24].

The dried plant (1 g) was boiled at a high temperature (about 200 °C) with a concentration of 98% sulfuric acid, filtered to obtain the solution to be determined, and then the elemental content of the plant was determined. Nitrogen content was measured using a Kjeldahl nitrogen apparatus after adding sodium hydroxide, hypochlorite, and phenol to 1 mL of the solution to be measured. The phosphorus content was measured spectrophotometrically after adding metadadic acid and molybdic acid to 1 mL of the liquid to be measured. After adding 1 mL of dilute hydrochloric acid to the 1 mL of liquid to be measured, the amount of potassium was determined by flame photometry.

### 2.3. Statistical Analysis

All data were sorted and analyzed using Excel2010, SPSS27.0, and Origin2018. All data are presented as mean ± standard deviation. Univariate analysis of variance (ANOVA) was used. Different letters in the figures indicate significant differences (* *p* < 0.05; ** *p* < 0.01).

## 3. Results and Discussion

### 3.1. Effect of Graphene on Soil after Leaching

#### 3.1.1. Effect of Graphene on Ion Concentration

The zeta potential of soil colloids showed an increasing trend with an increase in graphene concentration (Figure 1a). Detailed values are shown in Table 1. The zeta potential of the G20, G50, and G100 graphene-treated soil groups increased by 12.32%, 16.59%, and 17.49%, respectively, compared to that in the G0 control group. The zeta potential is the electrical potential of the sliding surface of the electric double layer of colloidal particles, which is located in the diffusion layer near the Stern side [25,26], as illustrated in Figure 1b. In colloidal environments, the zeta potential is commonly used to measure changes in the surface charge of colloidal particles, and it is affected by the concentration of ions within the Stern layer [27]. Boltzmann’s law explains how the magnitude of the ion concentration affects the magnitude of the bilayer thickness (K-1); when the added ion concentration is high, it compresses the diffusion layer in the bilayer, decreasing its thickness. When graphene was added to soil, the zeta potential of the soil colloid increased, indicating that the diffusion layer thickness increased. The diffusion layer increased without adding additional nutrient ions because the ions carried by the soil colloids themselves pooled in the diffusion layer, increasing the ion concentration in the local colloids. From these results, it can be concluded that the addition of graphene to soil enhances the aggregation of ions in soil colloids, thus increasing the ion concentration in that part of the soil. This is related to the π-π bonding energy of graphene itself interacting with ions in solution. Both anions and cations in a solution can interact with π-bonds, allowing ions to adsorb steadily on the graphene surface [28,29,30,31,32,33]. The dispersion of graphene in the soil colloidal solution also leads to an increase in ions in the colloidal solution.

#### 3.1.2. Effect of Graphene on Soil Nutrient Retention after Leaching

The soil leachate conductivity value of soil can reflect its salt content [34], and the lower the conductivity value, the lower the content of dielectric in the solution. Detailed values are shown in Table 2. As shown in Figure 2a, the conductivity of the soil leachate obtained solution showed a decreasing trend from G0 to G50, before increasing with an increase in graphene concentration at G100. The conductivity value of the drenching solution of each of the graphene-treated groups was lower than that of the control group, and the reduction in conductivity was more significant when the graphene concentration was lower than 100 mg L^−1^. The content of specific nutrient elements in the soil were measured to further investigate the retention of soil nutrient ions after leaching. As shown in Figure 2b–d, the retention of nutrients in the graphene-treated soils (G20, G50, G100) was higher than that in the control group (G0) after drenching. Among the nutrients, the content of ammonium nitrogen in the soil showed an increasing trend after drenching, and the retention of nitrogen in the soil increased with an increase in graphene concentration. The retention of ammonium nitrogen in G20, G50, and G100 samples increased by 16.59%, 21.06%, and 7.8%, respectively, compared with that in G0, and the G50 treatment exhibited the most evident effect on the retention of nitrogen. At the end of drenching, the effective phosphorus retention of the graphene-treated soil groups was more different from that of the control group (all values were higher than that of the control group). The retention of phosphorus was 53.62%, 38.20%, and 47.56% higher than that of the control group when the graphene concentration was 20, 50, and 100 mg/L, respectively. This indicated a more evident adsorption ability of phosphorus on graphene, which could effectively prevent the loss of phosphorus with the flow of water. The trend in content of quick-acting potassium in the soil after drenching was consistent with effective phosphorus, exhibiting 16.16%, 11.98%, and 11.57% higher content levels in the G20, G50, and G100 groups than in the control group, respectively.

The use of carbon nanomaterials as additives for soil fertilizer retention has been widely studied [35,36]. The addition of nanocarbon or biochar to soil can reduce nutrient loss and help retain nutrient ions in the soil because of the large specific surface area and strong adsorption of ions. After several wettings, the soil containing graphene sols reduced the loss of N, P, and K [20]. Under heavy rainfall conditions, the presence of nanocarbon can reduce nutrient loss from loess slopes, with less nutrient loss as the nanocarbon content is increased [35]. Graphene has a large specific surface area and adsorption capacity owing to its excellent structural properties. In the present study, the conductivity of the graphene-treated soil leachate after drenching was lower than that of the control group, indicating fewer electrolytes lost to the leachate solution and, presumably, greater nutrient ion retention in the soil. After drenching, the content of N, P, and K in the drenched soil of the graphene group was higher than that of the control group, indicating that graphene could enhance the fertilizer retention effect in the soil. Graphene can absorb PO₄^3^⁻ and reduce the leaching of NO^−^_3_ and NH^+^_4_ from soil solutions compared to conventional soil amendments [37]. In the present study, graphene has a better ability to absorb and accommodate elemental P, which is consistent with previous findings. In summary, the addition of graphene to soil can play a role in reducing nutrient loss because of the large specific surface area of graphene. This allows strong adsorption and enhanced retention of N, P, and K; prevents soil nutrient loss due to irrigation and rainfall; and improves soil fertility.

#### 3.1.3. Effect of Graphene on the Stability of Soil Microagglomerates

The stability of soil aggregates is one of the most important factors affecting soil fertility. The MWD and GMD are commonly used to determine the stability of soil aggregates. Larger values of both indicate a higher degree of soil agglomeration and greater stability and resistance to water erosion [38,39,40]. Soil particles <0.25 mm are referred to as soil microagglomerates; the soil selected for this experiment was a sandy soil with a large percentage of particles in the size range of 0.25–0.01 mm. The effect of graphene on soil microagglomerates can be seen in Figure 3a,b; the MWD and GMD of the soil Increased after the addition of graphene. MWD increased by 26.2%, 28.6%, and 31.7% in G20, G50, and G100 compared to that in G0, respectively, and GMD increased by 21.8%, 21.8%, and 26.1%, respectively. Detailed values are shown in Table 3. This indicates that the addition of graphene to the soil can change the stability of soil microagglomerates.

The addition of nanocarbon to soil increases the content of hydrostable aggregates and improves the soil structure, thus promoting soil stability [41]. Soil agglomerates are formed mainly by the electrostatic interaction of multivalent cations that bind organic and inorganic colloids in the soil [42]. Soil agglomerates are common inorganic colloids, and the π-π bonds in the graphene structure, with their π-electron clouds, can interact with the cations, causing them to adsorb around the graphene [26,43]. The addition of graphene allows the retention of cations in the soil, thus promoting the adhesion of soil colloids and improving their stability. Graphene material has a large specific surface area and strong adsorption properties, and it enters soil pores and adsorbs on soil particles, improving the degree of soil agglomeration.

### 3.2. Effect of Graphene on Plant Growth

The effects of different concentrations of graphene added to the soil on the height of pepper plants are shown in Figure 4a,e. Detailed values are shown in Table 4. After 60 days of cultivation, the plant height showed an increasing trend before slowly decreasing. The average height of all of the seedlings in the graphene treatment groups was greater than that in the control group (10.5%, 7.9%, and 2.6% greater in G20, G50, and G100 than in G0, respectively). The tallest pepper seedlings were found in the G20 group and the shortest in the G0 group. The level of nutrient uptake in plants is generally directly related to plant growth. The content of nutrient elements in the pepper plants was examined to further investigate whether graphene addition to soil promotes plant growth and nutrient uptake. After 60 d of growing pepper plants from seeds with or without added graphene, five plants were randomly removed from each experimental group, and the nutrient content of the plants was measured. As shown in Figure 4b–d, the content of nitrogen, phosphorus, and potassium in the plants grown in graphene-treated soil was higher than that in the control group. The content of nitrogen in G20, G50, and G100 plants was 6.98%, 2.79%, and 2.10% higher than that in the G0 group, respectively. The increase in phosphorus content in the plants grown in graphene-treated soil was more considerable when compared with the control group, especially in the G20 group, which exhibited 2.8 times more phosphorus than that in the G0 group. The phosphorus content of plants in the G50 and G100 groups was 15% and 33.1% higher than the control group, respectively, indicating that there were differences in the promotion effect of different concentrations of graphene on the uptake of phosphorus by the plants. The low concentration of graphene had a better effect on plant phosphorus uptake than the high concentrations of graphene. Graphene also had a promoting effect on potassium uptake by the pepper plants. The potassium content in the G20, G50, and G100 plants increased by 38.47%, 6.39%, and 3.88%, respectively. Regarding phosphorus content, the lowest concentration of graphene had a better effect on promoting the uptake of potassium than those of the higher concentrations of graphene. The leaching experiments demonstrated that graphene can improve the fertilizer retention capacity of soil, and the fertilizer preserved in soil needs to be absorbed by plants. Based on the interrelationship between the two, a double curve of soil nutrient retention and plant uptake performance was drawn, as shown in Figure 4f,h. The information in the figure shows that the retention percentages of all three NPK nutrient elements in the soil are higher than the plant uptake percentages. The results showed that the retained soil nutrients could be used by plants.

In this study, the addition of graphene to the soil promoted the growth of pepper seedlings and improved the accumulation of nutrient elements. All tested graphene concentrations were beneficial to plant growth, while graphene below 100 mg/L showed a greater promotion effect on pepper. These results are consistent with previous studies that reported the promotion of seed germination, plant root growth, and seedling height by low concentrations of graphene [44,45,46]. Conversely, high concentrations of graphene inhibit plant growth, because its nanotoxicity causes plant cell tissue damage, increases reactive oxygen species, and disrupts the soil microbial environment [47,48]. The promotion effect of graphene on plants in the present study was mainly reflected in the enhancement of soil fertility retention. The leaching experiment showed that the soil without graphene had more soluble nutrient runoff after rain soaking and washing, whereas the soil with graphene had less nutrient loss from the soil, which was beneficial to nutrient uptake and utilization. In addition, the structure of the soil after leaching was changed. The addition of graphene to the soil improved the stability of soil agglomerate structure and microagglomerates, which enhanced the erosion resistance of the soil, thus improving soil fertility. Thus, graphene plays a role in fertility retention in soil. A previous study found that graphite nanoparticles reduced nitrate leaching, promoted leaf growth, and increased lettuce yield [21]. Wu et al. [49] found that nanocarbon could reduce the total nitrogen content of paddy water and prevent nitrogen loss through runoff, thus improving the uptake and utilization of nitrogen and increasing the yield of rice. In our study, graphene increased the height of pepper plants and promoted the content of N, P, and K in 60-day-old plants, most likely because graphene increased the ionic concentration of soil colloids. The interaction of ions and graphene π-bonds resulted in anions and cations accumulating around the graphene, thus reducing the loss of soil nutrients. The strong adsorption capacity led to the retention of ions in the soil. This enhanced soil fertility and facilitated the absorption and utilization of nutrients.

## 4. Conclusions

Based on the double electric layer of the soil, this study investigated the effects of graphene on soil nutrient ion concentrations through the measurement of colloidal zeta potentials and leaching experiments. Pepper plant experiments demonstrated that graphene-enhanced soil nutrients could be absorbed and used by the plants to promote growth. Based on our experimental results, the following conclusions were made:(1)The addition of graphene to soil colloids increased the zeta potential of the colloids, which increased with an increase in graphene concentration. According to the Derjaguin–Landau–Verway–Overbeek theory, and what we understand about double electric layers, when only graphene is added to soil without increasing the fertilizer, the zeta potential of soil colloids increases. This represents the increase in ion concentration, indicating that graphene can increase the ion concentration of soil colloids.(2)After rainwater drenching and soaking, soil with added graphene could effectively reduce the loss of ammonium nitrogen, effective phosphorus, and fast-acting potassium. In terms of soil structure, graphene improved the stability and erosion resistance of soil microaggregates and improved soil fertility retention.(3)When graphene was applied to soil alongside fertilizer, plant growth was promoted. This was especially true when the graphene concentration was 20 mg/L, in which the plant height, root length, and nutrient absorption of pepper seedlings were most significant. In summary, our findings show that graphene can effectively reduce nutrient loss and promote plant nutrient uptake.(4)As a new type of soil conditioner and fertilizer additive, graphene has a strong adsorption effect on nutrient ions owing to its large specific surface area. This can reduce the loss of nutrients from the soil through rainfall leaching. At the same time, graphene can improve the utilization rate of nutrients by reducing their loss, which is conducive to plant absorption and growth promotion.

This study provides new insights into the application of graphene in agriculture by incorporating graphene into soil to enhance plant growth and by using the bilayer theory to analyze the effect of graphene on soil nutrient enhancement, which is of great significance.

## Figures and Tables

**Figure 1 materials-16-02578-f001:**
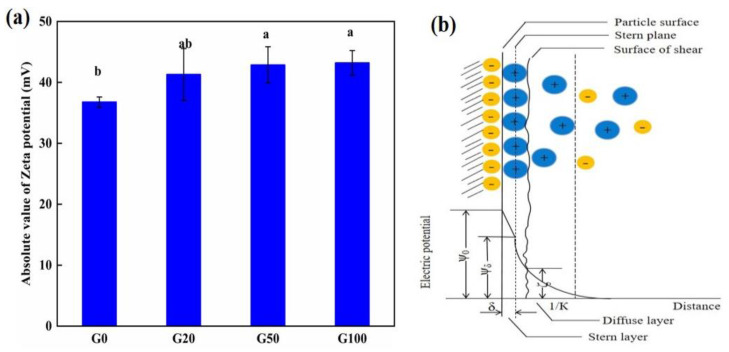
(**a**) Variation in absolute values of soil zeta potential with concentration of graphene. G0, G20, G50, and G100 indicate graphene concentration of 0, 20, 50, and 100 mg L^−1^, respectively. (**b**) Stern double electric layer model. The values represent the mean ± SD (n = 9). Different lowercase letters indicate significant differences at *p* < 0.05.

**Figure 2 materials-16-02578-f002:**
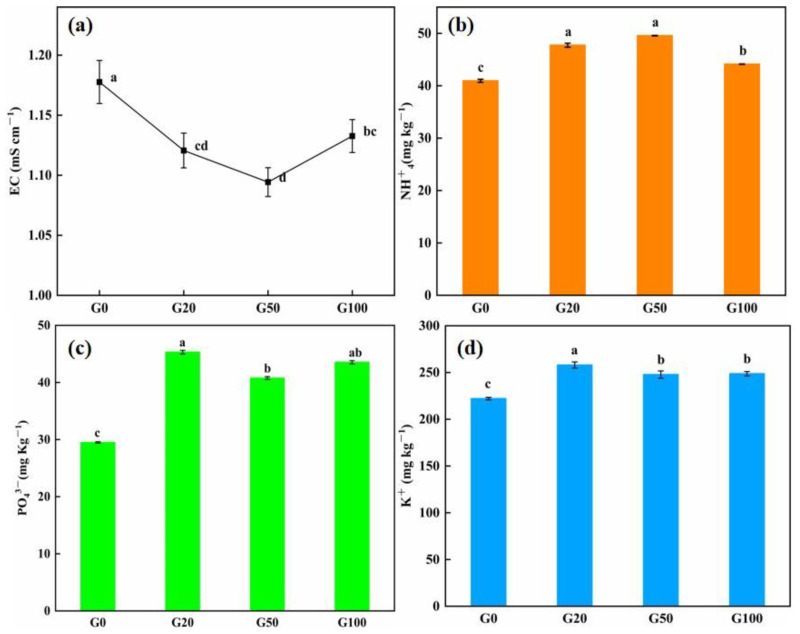
Effect of graphene on the content of the indicated element in soil after leaching. (**a**) Conductivity, and (**b**) NH^+^_4_, (**c**) PO₄^3^⁻, and (**d**) K^+^ content. Error bars represent the standard error of the mean (n = 5). The letters a, b, c, and d indicate statistically significant difference at *p* < 0.01 within each group comparison between the treatment.

**Figure 3 materials-16-02578-f003:**
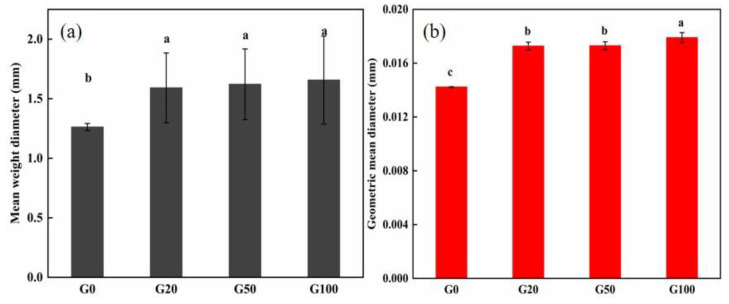
Effect of graphene on soil agglomerates. (**a**) Mean weight diameter (MWD) and (**b**) geometric mean diameter (GMD). Error bars represent the standard error of the mean (*n* = 5). The letters a, b, and c indicate a statistically significant difference at *p* < 0.05 within each group comparison between the treatment.

**Figure 4 materials-16-02578-f004:**
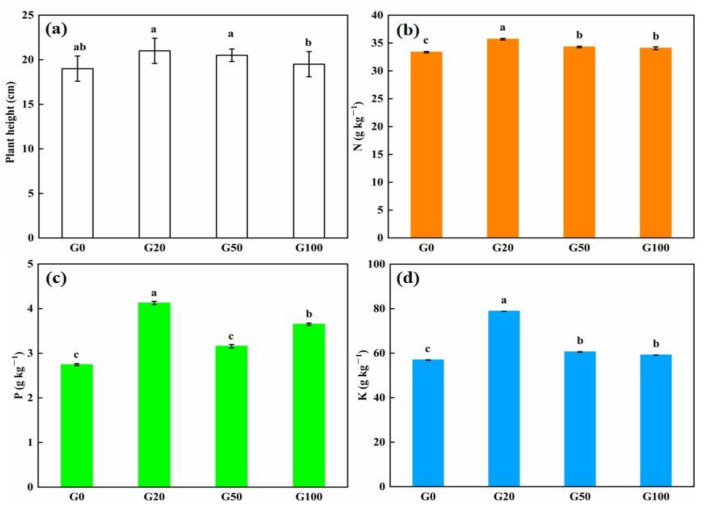
Effect of different concentrations of graphene added to the soil on the growth and nutrient uptake of pepper plants grown from seed for 60 days. (**a**) Plant height; (**b**) total nitrogen; (**c**) total phosphorus; (**d**) total potassium; (**e**) pepper plants after 60 days; (**f**) nitrogen biplot; (**g**) phosphorus biplot; (**h**) potassium biplot. Error bars represent the standard error of the mean (*n* = 5). The letters a, b, and c indicate a statistically significant difference at *p* < 0.05 within each group comparison between the treatment.

**Table 1 materials-16-02578-t001:** Variation in absolute values of soil zeta potential with concentration of graphene.

Group	G0	G20	G50	G100
Zeta (mV)	36.767 ± 0.839 b	41.300 ± 4.258 ab	42.867 ± 2.974 a	43.200 ± 2.022 a

Note: G0, G20, G50, and G100 represent those with and graphene at application rates of 0, 20, 50, and 100 mg/L, respectively. Error bars represent the standard error of the mean (n = 9). The letters a and b indicate statistically significant difference at *p* < 0.05 within each group comparison between the treatment.

**Table 2 materials-16-02578-t002:** Conductivity values and nutrient content of the eluent of drenched graphene-treated soil.

Group	G0	G20	G50	G100
EC (mS/cm)	1.178 ± 0.018 a	1.121 ± 0.014 ab	1.094 ± 0.012 d	1.133 ± 0.014 bc
NH^+^_4_ (mg/kg)	40.940 ± 0.291 c	47.733 ± 0.376 a	49.5633 ± 0.045 a	44.1267 ± 0.042 b
PO_4_^3^⁻ (mg/kg)	29.500 ± 0.125 c	45.320 ± 0.314 a	40.77 ± 0.251 b	43.53 ± 0.283 ab
K^+^ (mg/kg)	222.100 ± 1.345 c	258.000 ± 3.345 a	247.8 ± 3.897 b	248.700 ± 2.300 b

Note: G0, G20, G50, and G100 represent those with and graphene at application rates of 0, 20, 50, and 100 mg/L, respectively. Error bars represent the standard error of the mean (n = 5). The letters a, b, c, and d indicate statistically significant difference at *p* < 0.01 within each group comparison between the treatment.

**Table 3 materials-16-02578-t003:** Effect of graphene on soil agglomerates.

Group	G0	G20	G50	G100
MWD (mm)	1.262 ± 0.030 b	1.591 ± 0.293 a	1.621 ± 0.297 a	1.657 ± 0.370 a
GMD (mm)	0.142 ± 3.04 × 10^−5^ c	0.0173 ± 2.93 × 10^−4^ b	0.0173 ± 2.97 × 10^−4^ b	0.179 ± 3.69 × 10^−4^ a

Note: G0, G20, G50, and G100 represent those with and graphene at application rates of 0, 20, 50, and 100 mg/L, respectively. Error bars represent the standard error of the mean (*n* = 5). The letters a, b, and c indicate a statistically significant difference at *p* < 0.05 within each group comparison between the treatment.

**Table 4 materials-16-02578-t004:** Effect of different concentrations of graphene added to the soil on growth and nutrient uptake of pepper plants after 60 days grown from seeds.

Group	G0	G20	G50	G100
Plant height (cm)	19.0 ± 1.414 ab	21.0 ± 1.415 a	20.5 ± 0.707 a	19.5 ± 1.410 b
N (g/kg)	33.360 ± 0.125 c	35.690 ± 0.146 a	34.290 ± 0.134 b	34.06 ± 0.264 b
P (g/kg)	2.743 ± 0.024 c	4.125 ± 0.035 a	3.155 ± 0.040 c	3.650 ± 0.025 b
K (g/kg)	56.930 ± 0.158 c	78.830 ± 0.080 a	60.570 ± 0.193 b	59.140 ± 0.095 b

Note: G0, G20, G50, and G100 represent those with and graphene at application rates of 0, 20, 50, and 100 mg/L, respectively. Error bars represent the standard error of the mean (*n* = 5). The letters a, b, and c indicate statistically significant difference at *p* < 0.05 within each group comparison between the treatment.

## Data Availability

Not applicable.

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
