# Peer review of "Fertilizer-Holding Performance of Graphene on Soil Colloids Based on Double Electric Layer Theory"

_materials, 2023, doi:10.3390/ma16072578_

Round 1

Reviewer 1 Report

Journal: Materials

Title: Fertilizer-holding performance of graphene on soil colloids based on double electric layer theory

Author: Ziyan Liu , Ming Zhou, Wufang Liao , Jiayi Liu , Chaogui Luo , Chunyan Lu , Zhiwen Chen , Hongwei Zhu

Manuscript ID: materials-2172726

This paper focuses on influence of graphene presence on the promoting plants growth and nutrients. There are many factors that the authors have analyzed in this work, however I consider that some points should be modified or clarified

For this reason, my recommendation is to accept the paper for publication after a minor revision.

The introduction is very complete and well written makes clear what is the objective of the research and the previous work that has been carried out on the subject.

Materials and instruments: It should be commented why this soil was chosen and if it was related to the properties they describe. It would be interesting to know how these properties can influence the subsequent process.

Test method: Why are samples tested 9 times? Were different samples taken or was it the same measurement several times?

Pepper seedling cultivation and testing: I consider the experimental procedures to be very well described

Effect of graphene on ion concentration: Has this effect been observed with any other type of additive?

Figure 1: Taking into account the error is not very clear the increase and less between 50 and 100 to what could be due?

The results in all cases should go in a table although they are then explained in the text, group them

A large part of the explanation and bibliography of the results should be incorporated into the introduction.

Figure 2a: What is the reason why in the value of 100 it rises again?

Figure 2b) and d), Figure 3: There is no dependence on graphene content, what is the reason?

rewrite the conclusions. It is not necessary to repeat the values obtained although it is necessary to explain the reason

Author Response

Thank you so much for the review and please check the attached file.

Reviewer 2 Report

Overall, the manuscript entitled "Fertilizer-holding performance of graphene on soil colloids based on double electric layer theory" is an interesting study and good for the readership. However, there are comments that need to be addressed before its publication. These comments are as below:

General: English of the manuscript is not up to the level. For example, title is not well written and it must be revised.

Specific Comments:

1. Although authors separated soil colloids using sedimentation tank, it is difficult to ensure 2 micro-meter soil colloidal size.  

2. Why grephene was added at the rate of 0, 20, 50, and 100 ppm? Suitable ratios of the grephene needed to be pre-determined and then would have used for the study.

3. It would have been good if authors coated fertilizer with graphene to minimize losses and thereby enhancing nutrient availability in soil-plant systems.

4. Please describe the relevant material under sub-heading like characterization of soil ion composition instead of graphene mixing rates and experimentation.

5. Double layer data are often presented in curves. Authors are suggested to included curves for nutrient holding performance and their availability to the plants.

6. Test procedures for minerals are missing. Please check.

7. How did you measure ammonium from soil colloids?  

Reviewer 3 Report

The paper entitled “Fertilizer-holding performance of graphene on soil colloids based on double electric layer theory” by Ziyan Liu et al., in the current version cannot be published in materials. It presents very generic information and there is very little experimental data to support the claims of the authors.

Furthermore, the authors need to clarify:

1.     In line 37 the phrase "In nature, when contact is made between a liquid and a solid, the surface of the solid is electrically charged…"   is devoid of scientific meaning unless clarified.

2.     The authors should explain on line 113 what they mean by the term "graphene aqueous solutions", since graphene is not soluble in water.

3.     The authors center all their discussion on the results of figs 1, 2, 3. A the columns in the histogram do not show significant variations when compared with the experimental error of the determination.

4.     I don't think it's the measure so sensitive as to discriminate between 28.2 and 28.6, for example.

5.     On line 225 is the value 2.11 to be changed to 21.1 ?

6.     Figures 1, 2, 3 are illegible

7.     Between figures 3 and 4 there is an unnumbered figure.

8.     The authors describe the results of zeta potential and geometric mean diameter, which are macroscopic quantities. Furthermore, the GWD quantity refers to the hydrodynamic size and not to the size of the aggregate. Therefore, how can the authors describe the "bond" between graphene and fertilizer in mechanistic terms without further information?

Author Response

(The authors gave the same response as above.)

Round 2

Reviewer 2 Report

Dear Authors,

Although revised manuscript is considerably improved, authors are advised to improve their draft again in the the light of my points # 5 and 6 in earlier version.

Usually, curves are used for the retention of the nutrients or elements.

Also methodology of NPK determination on soils colloids needs to be included. Add brief details of of ammonical-nitrogen in methodology section too. 

Reviewer 3 Report

The corrections made by the authors are minimal compared to the observations made in the first review so that I cannot make further revisions.
